# The Obstetrician’s Role in Pregnant Women’s Decision-Making Process Regarding Influenza and COVID-19 Vaccination

**DOI:** 10.3390/vaccines11101608

**Published:** 2023-10-18

**Authors:** Agnieszka Sienicka, Agata Pisula, Katarzyna Karina Pawlik, Agnieszka Dobrowolska-Redo, Joanna Kacperczyk-Bartnik, Ewa Romejko-Wolniewicz

**Affiliations:** 1Students’ Scientific Group Affiliated to 2nd Department of Obstetrics and Gynecology, Medical University of Warsaw, 00-315 Warsaw, Poland; sienicka.aa@gmail.com (A.S.); katarzynapawlik97@gmail.com (K.K.P.); 22nd Department of Obstetrics and Gynecology, Medical University of Warsaw, 00-315 Warsaw, Poland; agnieszka.dobrowolskaredo@gmail.com (A.D.-R.); asiakacperczyk@gmail.com (J.K.-B.); ewamariar@wp.pl (E.R.-W.)

**Keywords:** influenza, COVID-19, vaccination, obstetrician, pregnancy

## Abstract

Pregnant women are considered to be a population vulnerable to influenza and COVID-19 infections, and the latest guidelines consistently recommend that they receive influenza and COVID-19 vaccinations. A cross-sectional questionnaire-based study was conducted among pregnant women in Poland to determine which factors have the greatest impact on their decision to vaccinate against influenza and COVID-19. A total of 515 pregnant women participated in the study. Among them, 38.4% (n = 198) demonstrated a positive attitude toward influenza vaccination, and 64.3% (n = 331) demonstrated a positive attitude toward COVID-19 vaccination. Logistic regression analysis revealed that the strongest influence on positive attitudes toward COVID-19 vaccination is having it recommended by an obstetrician–gynecologist (OR = 2.439, *p* = 0.025). The obstetrician–gynecologist’s recommendation to vaccinate against influenza also significantly influences the decision to vaccinate (OR = 5.323). The study results also show a strong correlation between the obstetrician–gynecologist as a source of information on influenza and vaccination and participants’ positive attitudes toward vaccination (OR = 4.163). Obstetricians have a significant influence on pregnant women’s decisions regarding vaccinations. Further recommendations to vaccinate and awareness-raising among obstetricians may be needed to increase the vaccination rate of pregnant women in Poland.

## 1. Introduction

Pregnant women are considered to be a vulnerable population due to their increased susceptibility to severe illness and adverse pregnancy outcomes associated with influenza and COVID-19 infection. The decision to receive vaccines against COVID-19 and influenza during pregnancy is a complex one that requires careful consideration of the risks and benefits. Pregnant women have various concerns surrounding vaccines, including their safety, efficacy, and potential side effects [1]. According to recommendations from experts of the Polish Society of Gynecologists and Obstetricians, pregnant individuals are advised to consider getting vaccinated against COVID-19 and influenza. The vaccinations can be administered at any point during pregnancy, with the preferable timing being the second or third trimester, following the phase of fetal organ formation. The use of an inactivated influenza vaccine is considered safe for both the mother and the unborn child. At present, there is no elevated risk associated with COVID-19 vaccination for pregnant individuals when compared to others of reproductive age. Moreover, there are no available data suggesting that the vaccine has any detrimental effects on fetal development starting from fertilization. However, due to the limitations in data needed to assess these outcomes according to evidence-based medicine standards, it is recommended that pregnant individuals consult their overseeing obstetrician before making a decision regarding vaccination [2,3].

The spread of misinformation and conspiracies about vaccines has led to vaccine hesitancy among certain individuals and communities. This hesitation can have significant consequences, such as outbreaks of vaccine-preventable diseases and a decrease in overall public health. It is crucial to address these concerns and provide accurate information to increase vaccine uptake and protect public health. An individual’s knowledge, attitudes, and beliefs (KAB) regarding vaccines are closely linked to their vaccination behavior [4]. The perinatal period, encompassing both the antenatal and postnatal phases, represents a critical window during which women frequently seek information regarding vaccinations [5]. The KAB they form throughout this time frame can have a profound impact on their vaccination behavior, including their adherence to vaccination recommendations during pregnancy and the likelihood of vaccinating their offspring [6]. Diverse sources of information about vaccination abound, from medical practitioners, including obstetricians, nurses, and general practitioners, to family and friends. The digital sphere, in the form of social media platforms and targeted vaccination campaigns, assumes a pivotal role in disseminating information on vaccines alongside traditional media sources like radio and television broadcasts. However, it is important to note that not all sources of information are equally reliable or accurate.

This study’s main goal was to understand how much influence obstetricians have on pregnant women’s decisions regarding COVID-19 and influenza vaccines. We wanted to determine what factors, such as medical advice, personal beliefs, and outside opinions, play a role in this important choice. We hope to contribute substantively to the ongoing discourse surrounding vaccine acceptance and its implications for public health, particularly in the context of pregnancy.

## 2. Materials and Methods

A cross-sectional questionnaire-based study was conducted among pregnant women in Poland to determine which factors have the greatest impact on their decision to vaccinate during the COVID-19 pandemic. The voluntary self-administered online survey was distributed to 92 Polish Facebook groups dedicated to women, mothers, or pregnant women. The questionnaire was created using the Google Forms survey administration software. The 92 Polish Facebook groups were selected because they were specifically tailored to the unique interests, concerns, and discussions of women, mothers, and pregnant women. The selection process involved a meticulous review of numerous Facebook groups, considering factors such as group size, engagement level, and the relevance of discussions to the target demographic. Only those groups that exhibited a substantial and active membership base and were actively discussing topics relevant to women’s experiences, motherhood, and pregnancy were included in the study. Furthermore, the research team ensured that the selected groups represented a diverse cross-section of the online community, encompassing a wide spectrum of interests, backgrounds, and experiences within the realm of women’s health and motherhood. This meticulous selection process was vital to ensure the survey findings would be robust, representative, and reflective of the broader population of Polish pregnant women engaging in these online social communities. Once the groups were identified and confirmed for inclusion, the survey was made available to their members, which provided a rich and diverse pool of potential respondents, each contributing valuable insights to the research endeavor. Confidentiality and anonymity were ensured. The data were collected from 24 October to 9 November 2021.

The developed questionnaire was subdivided into several sections and included 50 questions, both single-choice and multiple-choice. Basic sociodemographic and economic data, including age, educational level, marital status, place of residence, and average income per household member, were collected in the first part. The participant’s obstetric history and details of the current pregnancy were also collected. The emphasis was on gaining information regarding participants’ knowledge about influenza and COVID-19 and their vaccines, the sources of that knowledge, and the factors influencing vaccination and non-vaccination. The participants were also asked to provide their vaccination status for COVID-19 and influenza, along with their actual and preferred vaccination information sources. The entire questionnaire is available in Appendix A.

In order to achieve a 95% confidence level and a 5% margin of error, the sample size of the pregnant population was estimated at 385 participants.

All of the questionnaires were properly completed. The obtained data were analyzed using descriptive statistics in Microsoft Excel and univariate and multivariate logistic regression analyses for the categorical variables. Variables with a *p*-value less than 0.1 in the univariate regression analysis were included in the multiple regression model. The results were considered statistically significant if the p-value was less than 0.05. Statistical analysis of the data was performed using Statistica 12 software.

## 3. Results

The study included a total of 515 women between the ages of 19 and 43 years old. All of the answers were completed properly and used for further analysis. Table 1 presents the detailed sociodemographic factors of all participants. More than half of them were aged 26 to 35 years old (n = 387, 75.1%); 43.3% (n = 223) were between 26 and 30 years old and 31.8% (n = 164) were between 31 and 35 years old. Among the participants, 83.7% (n = 431) had received higher education, and 40.6% (n = 209) lived in a big city with over 500,000 residents. During the study, 11.7% of the women (n = 60) were in the first trimester of pregnancy, 38.1% (n = 196) were in the second trimester, and 50.3% (n = 259) were in the third trimester. Almost half of the surveyed women said this was their first pregnancy (n = 255, 49.5%), and 60% (n = 309) said this would be their first labor. The vast majority of the women reported that their current pregnancy was a single one (n = 508, 98.6%).

Only 21% of the participants (n = 108) had been vaccinated against influenza during their current pregnancy, and 17.5% (n = 90) intended to get vaccinated. Those two answers indicated a positive attitude toward influenza vaccination (n = 198, 38.4%). On the other hand, 79 women (15.3%) did not yet know if they would get vaccinated, and 238 (46.2%) did not intend to get vaccinated, which was cumulatively recognized as indicating a negative attitude toward this vaccination (n = 317, 61.6%).

Regarding vaccination against COVID-19, 64.3% of the participants (n = 331) showed a positive attitude; 154 (29.9%) had been vaccinated before their pregnancy, 145 (28.2%) during pregnancy, and 32 (6.2%) intended to get vaccinated. Among the responses, 35.7% (n = 184) indicated a negative attitude toward vaccination; 120 (23.3%) of the participants had not been vaccinated and did not intend to get vaccinated, and 64 (12.4%) were unsure about getting a COVID-19 vaccination.

Among the participants who demonstrated a positive attitude toward influenza vaccination (n = 198), 67.7% (n = 134) were offered vaccination; 95 (70.9%) received a recommendation from their obstetrician–gynecologist, 38 (28.4%) from another specialist, 21 (15.7%) from a nurse or midwife, and 35 (26.1%) were encouraged by family or friends. Regression analysis showed that the greatest influence on a positive attitude toward influenza vaccination was the encouragement of family or friends (OR = 10.626), followed by a recommendation by an obstetrician–gynecologist (OR = 5.323), another specialist (OR = 3.164), or a nurse or midwife (OR = 3.043) (Table 2).

Among the participants who were recommended to get an influenza vaccination by their obstetrician–gynecologist (n = 142), the majority (n = 95, 67%) received the vaccine or intended to do so. Those who had a negative attitude toward influenza vaccination (n = 317) were mostly not offered a vaccination (n = 244, 77%).

Among the respondents who had been vaccinated against COVID-19 during pregnancy or intended to be vaccinated (n = 177), 58.8% (n = 104) were recommended by their obstetrician–gynecologist, 26.6% (n = 47) by another specialist, 13% (n = 23) by a midwife or nurse, 18.6% (n = 33) were encouraged by their friends or family, and 28.8% (n = 51) were not offered a vaccination. Logistic regression analysis revealed that the strongest influence on a positive attitude toward COVID-19 vaccination was being recommended by an obstetrician–gynecologist (OR = 2.439, *p* = 0.025). Recommendations by another specialist or family/friends also had an effect on positive attitudes but to a lesser extent (Table 3).

Participants who indicated a negative attitude toward COVID-19 vaccination (n = 184) were mostly not offered vaccination (n = 113, 61.4%).

Participants were also asked about their actual and preferred sources of information about influenza and influenza vaccination. Among the women, 81.2% (n = 418) responded that they would prefer to obtain detailed information on influenza and vaccination from their obstetrician–gynecologist. However, only 20.6% of them (n = 106) indicated that an obstetrician–gynecologist was their actual source of information. The Internet and social media were identified as the main knowledge sources (n = 192, 37.3%), while 43.3% of the women (n = 223) did not search for information on influenza and vaccination.

Out of 106 respondents who said that an obstetrician–gynecologist was their source of information about influenza, 81 (76.4%) had a positive attitude about the influenza vaccination. Univariate and multivariate logistic regression analysis showed a strong correlation between the obstetrician–gynecologist as a source of information and participants’ positive attitude toward influenza vaccination (OR = 4.163) (Table 4 and Table 5). The area under the ROC curve was 0.739 ± 0.0224.

Moreover, participants who obtained information about influenza from their obstetrician–gynecologist were more likely to assess their level of knowledge as sufficient to make a conscious decision about influenza vaccination (OR = 3.082) (Table 6).

Interestingly, respondents who said that an obstetrician–gynecologist was their source of information about influenza demonstrated a greater level of knowledge about influenza vaccination and the course and complications of influenza. They were more likely to understand that pregnant women and infants are at increased risk of severe influenza and post-influenza complications (OR = 5.150 and 10.411, respectively) (Table 7 and Table 8).

Logistic regression analysis also showed a correlation between the obstetrician–gynecologist as a source of knowledge about influenza and the correct choice of recommended vaccinations for pregnant women in Poland, influenza, and pertussis (OR = 3.884) (Table 9).

## 4. Discussion

Pregnancy increases the risk of serious complications and hospitalization from seasonal influenza, and vaccination is an effective way to prevent them. The latest guidelines and vaccination programs consistently recommend influenza and COVID-19 vaccinations for pregnant women. Influenza vaccines can be administered at any time during pregnancy, preferably in the second or third trimester, since the data specifically show limited administration of influenza vaccines during the first trimester [2,7,8,9].

It is essential to discover tools that can increase the willingness of pregnant women to receive vaccinations, and many researchers worldwide wonder which factors impact their decision to vaccinate [10]. To address this, a team of researchers in the United States designed a personalized educational app called MomsTalkShots. The primary objective was to impact KAB by distributing readily available information customized to individuals based on their specific demographics and apprehensions. The use of MomsTalkShots led to an understanding of and attitudes around vaccines among pregnant women and mothers. Among women who were initially hesitant to vaccinate, the app increased their understanding of the likelihood of maternal influenza infection and enhanced their trust in the effectiveness of the influenza vaccine. For women who were unsure about their intention to vaccinate their infants, MomsTalkShots effectively alleviated their safety concerns. The application is presently undergoing updates to incorporate information on vaccines throughout all stages of life [11].

In a study performed in Korea, the results were similar to ours. Factors associated with influenza vaccine acceptance were knowledge of the influenza vaccine and trust in healthcare providers. The results emphasize the importance of providing adequate education to pregnant women in order to enhance their awareness about vaccination [12].

Obstetricians have a significant influence on pregnant women’s decisions about vaccination. They are often the primary source of information and advice, and their guidance can help women make informed choices about their health and that of their unborn child. Obstetricians are fully cognizant of their role and the influence they have on the decision-making process of pregnant women. A survey of obstetricians conducted in the United States found that 94% of them believed that they have an impact on pregnant women’s decision to receive an influenza vaccination during pregnancy. However, among the same group, fewer obstetricians believed they had an influence on their patients’ vaccination decisions for their children (only 47%) [13]. Moreover, a study conducted in Australia showed that any recommendation for influenza vaccine during pregnancy, not only by an obstetrician but by any healthcare provider, was a predictor of vaccine uptake [14]. An interesting survey was conducted among pregnant women in Ecuador, in which out of the 4.3% of respondents who did not receive either a recommendation or an influenza vaccination offer reported having been vaccinated, and 73.9% out of the group of 520 unvaccinated respondents identified not receiving a recommendation/offer by a health provider as a barrier to vaccination [15].

Despite guidelines and studies suggesting that obstetricians have a critical role in pregnant women’s decision-making process about vaccination, recommending influenza vaccination still might not be a common practice in obstetricians’ offices. In our study, only 6.8% of women (n = 35) said that the obstetrician recommended influenza vaccination before the pregnancy, and only 27.6% (n = 142) said that the obstetrician recommended vaccination during pregnancy. Other studies showed similar results. In a study conducted in Italy, only 14.7% of the surveyed women reported receiving information about the influenza vaccine from their gynecologist [16]. In a study conducted in Beijing, China, 44.3% of pregnant women stated that they would agree to receive an influenza vaccination if it was recommended by the physician, whereas only 19.4% of the surveyed obstetricians said they were willing to recommend the vaccine [17].

On the other hand, surveys of obstetricians conducted in other countries show different results. In a survey of obstetricians in Germany, the majority of the respondents (95.4%) said that they recommended influenza vaccination during pregnancy, and the same in a study of members of the national network of OB-GYNs representative of the American College of Obstetricians and Gynecologists (97%) [18,19]. Another study performed in Canada also showed that most obstetricians–gynecologists (78%) recommended the influenza vaccine to all of their pregnant patients [20].

An interesting study of gynecologists and family physicians was conducted in Mexico. It showed insufficient awareness of the potential side effects of influenza infection in the mother or fetus and its overall importance. The drivers of such beliefs were not assessed; however, such lack of knowledge might be an important factor in an obstetrician’s decision to recommend vaccination. Further studies are necessary to assess the scale and causes of this issue better [21]. Studies show that there might be a connection between the willingness to receive an influenza vaccination and the willingness to receive a SARS-CoV-2 vaccination. A survey of prenatal care providers in France showed that 49.4% of the participants, including 58.8% of obstetricians and gynecologists, would offer vaccination against SARS-CoV-2 to pregnant women, indicating that being used to prescribing seasonal influenza vaccines or supporting the vaccine for oneself improved vaccine prescription intention [22]. In a study from Switzerland, having had an influenza vaccine in the past year was a positive predictor for accepting a SARS-CoV-2 vaccine, and among the pregnant participants, those who had an obstetrician following their pregnancy were more likely to be willing to receive the SARS-CoV-2 vaccine [23].

In a previously mentioned study conducted in Korea, receiving a SARS-CoV-2 vaccine was significantly associated with accepting influenza vaccine among pregnant women. Nevertheless, the majority of participants who had a history of influenza vaccination reported that the COVID-19 pandemic did not affect their acceptance of the influenza vaccine or increase its importance [12]. Interestingly, in our study, there was also an evident correlation between COVID-19 and influenza vaccine uptake during pregnancy. However, more pregnant women received a recommendation for a SARS-CoV-2 vaccine (197; 38.3%) than an influenza vaccine (142; 27.6%) from their obstetrician.

Moreover, a recent survey showed that a limited number of obstetric practices provide pregnant women with details regarding routine childhood immunizations [24]. Unfortunately, prenatal visits are often overlooked as a potential opportunity to improve immunization rates among infants [6].

Further recommendations for vaccination and awareness-raising among obstetricians may be needed to increase their willingness to recommend influenza vaccination, particularly in countries where it is still not a common practice, especially since studies indicate that they have a key role in pregnant women’s decision to vaccinate.

During the peak of the COVID-19 pandemic, many women in Poland chose to postpone their plan to have a child, citing concerns related to the pandemic and limited access to healthcare facilities and medical personnel [25]. Our survey revealed that obstetricians play a vital role in influencing pregnant women regarding the choice to receive not only an influenza vaccine but also a COVID-19 vaccine. This finding is significant since a rise in the number of vaccinated pregnant women could help lower the spread of COVID-19 and mitigate fears associated with the pandemic. It could encourage women to resume their procreation plans, which could potentially improve the birth rate in Poland.

Also important is that the approval and support of public health workers are crucial for the success of vaccination programs, campaigns to promote vaccinations, and overall vaccine acceptance within society. In a survey of public health workers in Italy regarding the number of COVID-19 vaccine doses they received, the majority of participants had received a minimum of two doses. Among the 1000 individuals who took part in the study, only 5 had not received any vaccination. These individuals might have had medical reasons for not getting vaccinated, and they could probably obtain a negative swab test certificate if needed. These findings are promising because public health workers play a pivotal role as primary sources of vaccination information for the general public. Specifically, public health workers actively engage in promoting vaccinations across various healthcare settings and contribute to the development of effective communication tools and educational materials related to the vaccination process {Gallé, 2022 #195} [26].

The findings of our research should be considered within the context of certain constraints. The data did not come from a probability sample, so they may not be strictly representative of the population of interest, and the confidence intervals are not as meaningful as they would be from a probability sample. If the responses are clustered or correlated, the p-values may be artificially small, and the confidence intervals artificially narrow compared to what would be observed with a probability sample. Nevertheless, utilizing this data collection approach enabled us to gain insights about pregnant women from a variety of backgrounds and geographical areas in the country.

## 5. Conclusions

Pregnant women are considered to be a group susceptible to contracting influenza and COVID-19, and current guidelines consistently recommend that they be vaccinated against both of these diseases. The prevalence of expectant mothers indicating that they had been immunized or expressing their intention to be vaccinated against those diseases is not high enough. There are factors associated with their decision. Obstetricians seem to have a significant influence on pregnant women’s decisions regarding vaccinations. Further recommendations for vaccination and awareness-raising among obstetricians may be needed to increase the vaccination rate of pregnant women in Poland.

## Figures and Tables

**Table 1 vaccines-11-01608-t001:** Sociodemographic and obstetric characteristics of participants.

Category (n = 515)	Variables	Frequency	Percentage
Age	19–25	64	12.4%
	26–30	223	43.3%
	31–35	164	31.8%
	36–40	57	11.1%
	41–43	7	1.4%
Education	Primary	2	0.4%
	Vocational	5	1.0%
	Secondary	59	11.5%
	Studying	18	3.5%
	Higher	431	83.7%
Average income per household member	<1000 PLN	15	2.9%
	1000–2000 PLN	65	12.6%
	2000–3000 PLN	108	21.0%
	3000–4000 PLN	132	25.6%
	4000–5000 PLN	79	15.3%
	>5000 PLN	116	22.5%
Place of residence	Countryside	104	20.2%
	Small village (<50k residents)	68	13.2%
	Town (50k–100k residents)	34	6.6%
	City (100k–500k)	101	19.6%
	City (>500k)	209	40.6%
Current relationship status	Single	3	0.6%
	Informal relationship	96	18.6%
	Married	414	80.4%
	Divorced	2	0.4%
Week of gestation	1–13	60	11.7%
	14–27	196	38.1%
	28–40	259	50.3%
Number of previous pregnancies	0	255	49.5%
	1	167	32.4%
	2	63	12.2%
	3+	30	5.8%
Number of previous labors	0	309	60.0%
	1	154	29.9%
	2	41	8.0%
	3+	11	2.1%
Current pregnancy	Single	508	98.6%
	Multiple	7	1.4%
Having children	Yes	204	39.6%
	No	311	60.4%

**Table 2 vaccines-11-01608-t002:** The association between the influenza vaccination recommender and getting vaccinated or willingness to vaccinate (univariate regression analysis).

		95% CI		
Variables	OR	Lower	Upper	*p*-Value
Obstetrician–gynecologist	5.323	3.423	8.277	<0.001
Other doctor	3.164	1.521	6.584	0.002
Nurse/midwife	3.043	1.045	8.866	0.041
Family/friends	10.626	4.163	27.117	<0.001

**Table 3 vaccines-11-01608-t003:** The association between the COVID-19 vaccination recommender and getting vaccinated or willingness to vaccinate (univariate regression analysis).

		95% CI		
Variables	OR	Lower	Upper	*p*-Value
Obstetrician–gynecologist	2.439	1.119	5.315	0.025
Other doctor	1.902	0.913	3.961	0.086
Nurse/midwife	0.771	0.350	1.697	0.518
Family/friends	1.290	0.626	2.660	0.490

**Table 4 vaccines-11-01608-t004:** The association between different sources of knowledge about influenza and influenza vaccination and positive attitudes toward vaccination (univariate regression analysis).

		95% CI		
Variables	OR	Lower	Upper	*p*-Value
Obstetrician–gynecologist	4.163	2.444	7.092	<0.001
Other doctor	1.256	0.758	2.079	0.376
Nurse/midwife	1.219	0.592	2.510	0.590
Family/friends	0.785	0.468	1.319	0.361
Social media/Internet	1.182	0.739	1.891	0.485
TV/Radio	0.318	0.130	0.778	0.012
Leaflets/campaigns/banners	0.627	0.351	1.120	0.115
No source	0.152	0.074	0.314	<0.001

**Table 5 vaccines-11-01608-t005:** The association between different sources of knowledge about influenza and influenza vaccination and positive attitudes toward vaccination (multivariate analysis).

		95% CI		
Variables	OR	Lower	Upper	*p*-Value
Obstetrician–gynecologist	4.617	2.752	7.746	<0.001
TV/radio	0.278	0.117	0.663	0.004
No source	0.151	0.082	0.278	<0.001

**Table 6 vaccines-11-01608-t006:** The association between different sources of knowledge about influenza and influenza vaccination and assessing the level of knowledge as sufficient to make a conscious decision on influenza vaccination (univariate regression analysis).

		95% CI		
Variables	OR	Lower	Upper	*p*-Value
Obstetrician–gynecologist	3.082	1.599	5.942	0.001
Other doctor	2.775	1.523	5.055	0.001
Nurse/midwife	2.067	0.810	5.273	0.129
Family/friends	0.598	0.351	1.021	0.060
Social media/ Internet	1.695	1.018	2.825	0.043
TV/radio	0.569	0.269	1.204	0.140
Leaflets/campaigns/banners	0.845	0.473	1.511	0.570
No source	0.440	0.240	0.805	0.008

**Table 7 vaccines-11-01608-t007:** The association between different sources of knowledge about influenza and influenza vaccination and the indication that pregnant women are at increased risk of severe influenza and post-influenza complications (univariate regression analysis).

		95% CI		
Variables	OR	Lower	Upper	*p*-Value
Obstetrician–gynecologist	5.150	1.954	13.576	0.001
Other doctor	0.858	0.446	1.651	0.646
Nurse/midwife	1.185	0.430	3.271	0.743
Family/friends	2.022	0.981	4.167	0.056
Social media/ Internet	1.256	0.681	2.318	0.465
TV/radio	0.691	0.289	1.651	0.405
Leaflets/campaigns/banners	0.749	0.384	1.462	0.397
No source	0.529	0.266	1.049	0.068

**Table 8 vaccines-11-01608-t008:** The association between different sources of knowledge about influenza and influenza vaccination and the indication that infants are at increased risk of severe influenza and post-influenza complications (univariate regression analysis).

		95% CI		
Variables	OR	Lower	Upper	*p*-Value
Obstetrician–gynecologist	10.411	2.412	44.936	0.002
Other doctor	1.142	0.531	2.457	0.733
Nurse/midwife	0.740	0.260	2.105	0.572
Family/friends	1.551	0.696	3.455	0.283
Social media/ Internet	1.924	0.931	3.972	0.077
TV/radio	1.195	0.379	3.766	0.762
Leaflets/campaigns/banners	0.980	0.445	2.159	0.960
No source	1.141	0.521	2.500	0.741

**Table 9 vaccines-11-01608-t009:** The association between different sources of knowledge about influenza and influenza vaccination and the correct choice of recommended vaccinations for pregnant women in Poland (univariate regression analysis).

		95% CI		
Variables	OR	Lower	Upper	*p*-Value
Obstetrician–gynecologist	3.884	1.854	8.136	<0.001
Other doctor	0.550	0.316	0.958	0.035
Nurse/midwife	0.615	0.284	1.331	0.217
Family/friends	0.715	0.414	1.237	0.231
Social media/Internet	1.469	0.857	2.516	0.162
TV/radio	0.262	0.123	0.556	<0.001
Leaflets/campaigns/banners	0.756	0.415	1.374	0.358
No source	0.214	0.115	0.400	<0.001

## Data Availability

All the data used in this study was presented in the article.

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
