# Peer review of "The Obstetrician’s Role in Pregnant Women’s Decision-Making Process Regarding Influenza and COVID-19 Vaccination"

_vaccines, 2023, doi:10.3390/vaccines11101608_

Round 1

Reviewer 1 Report

The manuscript is interesting and should be published after revisions.

The statistical methods are not well described and not clearly reported.

The paper needs a section that lists the limitations of the work.

Methods

State that the data were analyzed as if they came from a simple random sample of pregnant women.  In fact, they came from a sample who a) heard about the survey and b) had the ability and inclination to respond.  This sample may not be representative of all pregnant women. The sample may be clustered or correlated based on whether women who learned of the survey have factors in common that they do not share with women who did not hear of it, or who were not inclined or were unable to respond.  That clustering or correlation of responses may lead to an exaggerated appearance of statistical significance.

For multivariable logistic regression, state clearly:

a)       Which variables were considered to be put in the model

b)      What criteria were used to include them

c)       Whether any were dropped from the model (e.g., if their p-value was > 0.05 and if dropping them did not change the coefficients of the remaining terms by more than 10-20%)

d)      Did you consider or try adding any interaction terms in the model? 

e)      When you evaluated the goodness-of-fit of the final model, was it acceptable?

f)        What was the concordance statistic, or area under the ROC curve?  The reader will want to know how well the model distinguishes between 0s and 1s.

Results

Every table should indicate clearly whether it is reporting results of univariate or multivariable regression.

A p-value is never actually 0.000.  Those should be reported as < 0.001.

For univariate logistic regression, the odds ratio is difficult to interpret so it might be clearer to report risk ratios.  Of course, for multivariate regression, only the odds ratio is possible.

It is not correct, and too strong to label these results as “the impact” of X on Y.  In a cross-sectional survey you cannot infer causality.  You are reporting “associations” or “correlations” but it is not correct to say that you have measured “impact”.  Modify the language to acknowledge this limitation.

Limitations

Include a section saying that the data do not come from a probability sample, so the data are likely not strictly representative of the population of interest and the confidence intervals are not as meaningful as they would be from a probability sample.  If the responses are clustered or correlated, the p-values may be artificially small and the confidence intervals may be artificially narrow compared to what would be observed with a probability sample.

Author Response

Response to Reviewer 1 Comments

Point 1: Methods

State that the data were analyzed as if they came from a simple random sample of pregnant women.  In fact, they came from a sample who a) heard about the survey and b) had the ability and inclination to respond.  This sample may not be representative of all pregnant women. The sample may be clustered or correlated based on whether women who learned of the survey have factors in common that they do not share with women who did not hear of it, or who were not inclined or were unable to respond.  That clustering or correlation of responses may lead to an exaggerated appearance of statistical significance.

For multivariable logistic regression, state clearly:

  1. a) Which variables were considered to be put in the model
  2. b) What criteria were used to include them
  3. c) Whether any were dropped from the model (e.g., if their p-value was > 0.05 and if dropping them did not change the coefficients of the remaining terms by more than 10-20%)
  4. d) Did you consider or try adding any interaction terms in the model?
  5. e) When you evaluated the goodness-of-fit of the final model, was it acceptable?
  6. f) What was the concordance statistic, or area under the ROC curve? The reader will want to know how well the model distinguishes between 0s and 1s.

Response 1: Thank you very much for your suggestions. We have added suggested information regarding the study’s methodology in the manuscript.

Variables with a p-value <0.1 in the univariate regression analysis were included in the multiple regression model and no variables were dropped from the model. We didn’t consider adding any interaction terms in the model and the goodness-of-fit of the final was acceptable when evaluated. We have also added the value of the area under the ROC curve in the results section.

Point 2: Results

Every table should indicate clearly whether it is reporting results of univariate or multivariable regression.

A p-value is never actually 0.000.  Those should be reported as < 0.001.

For univariate logistic regression, the odds ratio is difficult to interpret so it might be clearer to report risk ratios.  Of course, for multivariate regression, only the odds ratio is possible.

It is not correct, and too strong to label these results as “the impact” of X on Y.  In a cross-sectional survey you cannot infer causality.  You are reporting “associations” or “correlations” but it is not correct to say that you have measured “impact”.  Modify the language to acknowledge this limitation.

Response 2: We have included in each table an information whether it is reporting results of univariate or multivariate regression analysis and replaced a p-value 0.000 with <0.001.

In our opinion, presenting only the odds ratio is sufficient. We did not want to report both values simultaneously so that the tables would not become unreadable to readers.

We have also modified the language- replaced the phrase “impact” with more appropriate terms.

Point 3: Limitations

Include a section saying that the data do not come from a probability sample, so the data are likely not strictly representative of the population of interest and the confidence intervals are not as meaningful as they would be from a probability sample.  If the responses are clustered or correlated, the p-values may be artificially small and the confidence intervals may be artificially narrow compared to what would be observed with a probability sample.

Response 3: We have included a section that lists the limitations of the study.

Reviewer 2 Report

This manuscript, by Sienicka, A et al., presented a cross-sectional, questionnaire-based study aimed to address the obstetrician’s role in the pregnant women’s decision-making process about influenza and COVID-19 vaccination in Poland. 

The study included 515 pregnant women, the conclusion of the study is that obstetricians have a significant influence on pregnant women’s decision about vaccination. 

The result makes sense and is expected.  The discussion section has touched several good points regarding the common practice in the US and some other countries.  One major shortcoming of the manuscript is the lack of discussion of the norm of the current Polish clinical practice regarding this issue.  In another word, with this finding, how much improvement of Polish obstetricians can do to encourage patients to get vaccinated?  What are the views/attitude of the obstetricians toward the vaccination in Poland?   

Author Response

Response to Reviewer 2 Comments

Point 1: One major shortcoming of the manuscript is the lack of discussion of the norm of the current Polish clinical practice regarding this issue.  In another word, with this finding, how much improvement of Polish obstetricians can do to encourage patients to get vaccinated?  What are the views/attitude of the obstetricians toward the vaccination in Poland?  

Response: Thank you very much for your comment. Only 20.6% (n=106) of respondents indicated that an obstetrician-gynaecologist was their actual source of information. This suggests that while the majority of obstetricians in Poland hold a favorable view towards vaccination, they do not adequately prioritize the task of educating patients about vaccination. With the knowledge of how their discussion with the patient can lead to higher vaccination rate, the obstetricians might prioritize it more.

As it comes to the obstetricians' attitude toward the vaccination in Poland, it is an interesting topic, however it needs to be researched more since we haven’t found such studies, and our study focuses more on the perspective from the patient's side. 
